# miR-215 Targeting Novel Genes *EREG*, *NIPAL*1 and *PTPRU* Regulates the Resistance to *E.coli* F18 in Piglets

**DOI:** 10.3390/genes11091053

**Published:** 2020-09-07

**Authors:** Chao-Hui Dai, Fang Wang, Shi-Qin Wang, Zheng-Chang Wu, Sheng-Long Wu, Wen-Bin Bao

**Affiliations:** 1Key Laboratory for Animal Genetics, Breeding, Reproduction and Molecular Design of Jiangsu Province, College of Animal Science and Technology, Yangzhou University, Yangzhou 225009, China; dx120170081@yzu.edu.cn (C.-H.D.); wangfangRD@163.com (F.W.); w17351373585@163.com (S.-Q.W.); zcwu@yzu.edu.cn (Z.-C.W.); slwu@yzu.edu.cn (S.-L.W.); 2Joint International Research Laboratory of Agriculture & Agri-Product Safety, Yangzhou University, Yangzhou 225009, China

**Keywords:** pig, miR-215, target genes, RNA-seq, promoter region, *E. coli* resistance

## Abstract

Previous research has revealed that miR-215 might be an important miRNA regulating weaned piglets’ resistance to *Escherichia coli* (*E. coli)* F18. In this study, target genes of miR-215 were identified by RNA-seq, bioinformatics analysis and dual luciferase detection. The relationship between target genes and *E. coli* infection was explored by RNAi technology, combined with *E. coli* stimulation and enzyme linked immunosorbent assay (ELISA) detection. Molecular regulating mechanisms of target genes expression were analyzed by methylation detection of promoter regions and dual luciferase activity assay of single nucleotide polymorphisms (SNPs) in core promoter regions. The results showed that miR-215 could target *EREG*, *NIPAL1* and *PTPRU* genes. Expression levels of three genes in porcine intestinal epithelial cells (IPEC-J2) in the RNAi group were significantly lower than those in the negative control pGMLV vector (pGMLV-NC) group after *E. coli* F18 stimulation, while cytokines levels of TNF-α and IL-1β in the RNAi group were significantly higher than in the pGMLV-NC group. Variant sites in the promoter region of three genes could affect their promoter activities. These results suggested that miR-215 could regulate weaned piglets’ resistance to *E. coli* F18 by targeting *EREG*, *NIPAL1* and *PTPRU* genes. This study is the first to annotate new biological functions of *EREG*, *NIPAL1* and *PTPRU* genes in pigs, and provides a new experimental basis and reference for the research of piglets disease-resistance breeding.

## 1. Introduction

MicroRNAs (miRNAs) are a class of conserved endogenous noncoding small RNAs with about 22 nt length. They are widely distributed in cells of plants, animals, nematodes and humans and play important roles in biological processes by regulating gene expression at the post-transcriptional level [1]. It has been found that miRNAs are widely involved in various organs’ development, cell proliferation and differentiation, migration and apoptosis, lipogenesis metabolism, insulin secretion and other physiological activities [2,3,4,5]. At the same time, they play important regulatory roles in pathological processes such as tumor, viral and bacterial diseases [6,7,8,9]. Recent research identified molecular networks within genes, and miRNAs were candidate factors determining adipogenesis, fatness, and selected fattening characteristics in pigs [10]. Among them, miR-215 has been found to be associated with tumors in the colon, liver, lung, kidney and cervix, and is viewed as a tumor suppressor miRNA [11,12,13,14,15,16]. There are also some reports on studies of miR-215 in pigs. The differential expression of miRNAs in Large White pigs and Meishan pigs was analyzed to investigate the regulation of miRNAs on adipogenesis [17], and the results showed that the expression of miR-215 in Large White pigs was significantly higher than that in Meishan pigs. Thus, miR-215 is likely to play an important regulatory role in the process of pork fat formation. Ten skeletal muscle tissues of extreme individuals in Yorkshire boars with high and low residual feed intake (RFI) were selected for differential expression analysis of miRNAs and the study revealed that miR-215 was down-regulated in low RFI pigs [18], which suggested that miR-215 also has a certain regulatory effect on pig feed efficiency. The expression of miR-215 in pig skeletal muscle was significantly down-regulated after injection of lipopolysaccharide [19], suggesting that miR-215 plays an important regulatory role in acute inflammation. Sharbati et al. [20] performed miRNA sequencing studies on the intestinal tissues including the duodenum, jejunum anterior segment, jejunum posterior segment, ileum, ascending colon and transverse colon of 31-day-old piglets and found that miR-215 was highly expressed in the duodenum and jejunum, suggesting that miR-215 may play an important role in intestinal diseases and individual immune processes.

In summary, miR-215 plays important regulatory roles in the pig growth development and immune process. In a previous study, we conducted the Illumina Solexa high-throughput sequencing technology to analyze differentially expressed microRNAs in the duodenum between *E. coli* F18 resistant and sensitive Sutai pigs, and found that the miR-215 expression level in the *E. coli* F18-sensitive group was significantly higher than that in the *E. coli* F18-resistant group [21]. Therefore, miR-215 may be an important miRNA that regulates piglets’ resistance to *E. coli* F18. However, the molecular mechanism of miR-215 regulating the resistance of piglets to *E. coli* F18 remains poorly understood.

In this study, a porcine intestinal epithelial cell line (IPEC-J2) with miR-215 stable silencing was constructed and transcriptome sequencing analysis was performed. Combined with bioinformatic analysis and dual luciferase activity assays, the target genes of miR-215 were predicted and identified. The regulation effect of miR-215 on weaned piglets’ resistance to *E. coli* F18 diarrhea was further analyzed by the method of gene interference technology (RNAi), bacterial stimulation and ELISA detection. Finally, the molecular regulation mechanisms of target genes’ expression were investigated by methylation analysis of cytosine guanine (CpG) islands and dual luciferase assay of SNPs in core promoter regions. This study aims to reveal the molecular mechanism of miR-215 regulation of resistance to *E. coli* F18 and to provide a scientific basis for functional genes and effective genetic markers of *E. coli* resistance.

## 2. Materials and Methods

### 2.1. Ethics Approval and Consent to Participate

Animal experiments were approved by the Institutional Animal Care and Use Committee of the Yangzhou University Animal Experiments Ethics Committee (permit number: SYXK [Su] IACUC 2012-0029). All experimental procedures were performed in accordance with the Regulations for the Administration of Affairs Concerning Experimental Animals approved by the State Council of the People’s Republic of China.

### 2.2. Animals

The experimental pigs used in this study included *E. coli* F18-resistant and sensitive Sutai pigs from the Sutai Pig Breeding Center in Suzhou (Jiangsu Province, China). We screened three resistant and three sensitive animals of similar birth weight, weaning weight, body shape and coat color using a V-type secretion system [22]. All pigs were housed in the same environment. At 35 days of age, when piglets are most susceptible to *E. coli* F18, the animals were sacrificed, and duodenum tissues were separated and transferred to liquid nitrogen.

### 2.3. Primer Design and Synthesis

According to the mature sequence of miR-215: AUGACCUAUGAAUUGACAGAC (GenBank accession number: NR_038518.1), real-time quantitative PCR (qPCR) primers of miR-215, reverse transcription kits and fluorescent quantitative detection kits were purchased from Tiangen Biotech Co., Ltd. (Beijing, China). U6 was used as a house-keeping gene to detect the transcript level of miR-215. The qPCR primers of differentially expressed genes (DEGs) from RNA-seq were designed using Primer3web (http://primer3.ut.ee/), and the *GAPDH* gene was used as a house-keeping gene to detect the transcript level of DEGs. The specific information of the primers is shown in Appendix A. All the primers for DEGs were synthesized by Sangon Biotech Co., Ltd. (Shanghai, China).

### 2.4. Expression Changes of miR-215 in IPEC-J2 Cells Stimulated by E. coli F18ac

IPEC-J2 cells were seeded into a 12-well plate at a density of 5.0 × 10^5^ per well and were cultured at 37 °C and 5% CO_2_ for 24 h, and each treatment group was conducted with three replicates. *E. coli* F18ac was used to stimulate IPEC-J2 cells according to our previous study [23]. Cells were extracted with an miRNA extraction kit (Tiangen Biotech Co., Ltd., Beijing, China) to analyze the expression level of miR-215. The corresponding reverse transcription kit and fluorescence quantitative detection kit (Beijing Tiangen Biotechnology Co., Ltd., Beijing, China) were ordered. cDNA synthesis and qPCR tests were performed strictly following the kit instructions. Each treatment was conducted with three replicates and the relative quantification results were treated by the 2^−ΔΔCt^ method [24]. U6 was used as a house-keeping gene, and the expression level of miR-215 in the no-stimulation group was defined as 1 to calculate the relative expression level of miR-215 in the *E.coli* F18ac stimulation group.

### 2.5. miR-215 Inhibitor Lentivirus Transfected IPEC-J2 Cell Line

IPEC-J2 cells were transferred into a 12-well plate at a density of 1 × 10^6^/well, and cultured in a DMEM-F12 medium (Gibco, Gaithersburg, MD, USA) containing 10% fetal bovine serum (FBS) (Gibco, Gaithersburg, MD, USA) in a 37 °C, 5% CO_2_ incubator. When the cell coverage rate was about 70%, the lentiviruses (GenePharma, Suzhou, China) of miR-215 inhibitor and the negative control (NC) were transfected into IPEC-J2 cells. Each treatment was conducted with three replicates. The cells were cultured overnight in a 37 °C, 5% CO_2_ incubator. After 24 h, the expression of green fluorescent protein (GFP) was observed and puromycin was added to screen the positive cells. After the positive cells were stably expressed, the cells were collected for quantitative detection to analyze the efficiency of miR-215 inhibition.

### 2.6. RNA-Seq Analysis

#### 2.6.1. Library Construction and Transcriptome Sequencing

The IPEC-J2 cells of the miR-215 inhibitor group and NC group were analyzed by RNA-seq, and three biological samples were included in the miR-215 inhibitor group L1, L2, L3 and NC group NC1, NC2 and NC3. Total cellular RNA was extracted using Trizol reagent (Invitrogen, Carlsbad, CA, USA) and the RNA samples were subjected to quality inspection. The mRNA of eukaryotes was then enriched by binding of A-T complementary pairing to the polyA tail of mRNA using magnetic beads with Oligo (dT). Subsequently, the fragmentation buffer was used to break the mRNA into short fragments and mRNA was used as a template to synthesize a strand of cDNA using random hexamers, and then double-stranded (ds) cDNA was synthesized by adding buffer, dNTPs and DNA polymerase I. The dscDNA was subsequently purified using AMPure XP beads. The purified dscDNA was subjected to terminal repair. A tail was added, and the sequencing linker was ligated, and then AMPure XP beads were used for fragment size selection. PCR enrichment was performed to obtain a final cDNA library. Finally, it was sequenced on the IlluminaHiseq 4000, and the amounts of data were about 6G clean data/sample.

#### 2.6.2. Bioinformatics Analysis and Quantitative Verification

Quality control of the raw reads data obtained by sequencing of Hiseq4000 was conducted using NGS QC Toolkit v2.3.3 software [25] to remove low quality fragment filtering and obtain clean reads (clean data were obtained by removing reads containing adapter, reads containing poly-N and low quality reads from raw data.), establishing a reference genome index using bowtie2 software [26], and Tophat2 software (http://tophat.cbcb.umd.edu/. The filtered reads were then compared to the reference genome. Differential expression analysis of two groups was performed using the DESeq R package (1.8.3) [27]. A series of subsequent analyses of the predicted mRNA sequences (including structural annotations and functional annotations) and gene expression were performed, and genes differentially expressed between the samples were screened from the gene expression results (|log_2_(Foldchange)| > 1 and padj < 0.05). Here, padj refers to p-adjusted that is corrected after multiple-test adjustment by the method of Benjamini & Hochberg. Gene Ontology (GO) seq based Wallenius noncentral hyper-geometric distribution [28] was implemented for GO enrichment analysis. KOBAS software [29] was used to test the statistical enrichment of the differentially expressed genes in Kyoto Encyclopedia of Genes and Genomes (KEGG) pathways. All the gene sets annotated to the Entrez database for significant difference analysis were used as backgrounds in this analysis. Cluster analysis, GO function significant enrichment analysis (padj < 0.05) and pathway significant enrichment analysis (padj < 0.05) were performed based on differentially expressed genes.

### 2.7. Prediction and Verification of Porcine miR-215 Target Genes

Combined with the results of differential gene screening, the online websites TargetScan (http://www.targetscan.org/vert_71/) and miRTarBase (http://mirtarbase.mbc.nctu.edu.tw/php/search.php) were used to predict the possible target genes of miR-215, and then verified by the dual luciferase reporter system. The pMIR-Report Luciferase vector was digested with *Mlu*I and *Hind*III, and the linear vector was recovered by 1% agarose gel electrophoresis. The oligo was annealed into double strands, recombined to the linearized pMIR-Report Luciferase vector by T4 DNA ligase. Recombined vectors were transformed into competent cells’ DH-5α, a monoclonal clone was picked, and the positive cloned recombinant plasmid vectors confirmed by sequencing were extracted and purified. 293T cells were cultured in Dulbecco’s modification of Eagle’s medium Dulbecco (DMEM) (Gibco, Gaithersburg, MD, USA) containing 10% FBS in culture dishes. After growth until the density reached 70%, recombined plasmids, PRL-TK plasmid and chemically synthesized miR-215 mimics and miR-215 inhibitor (GenePharma, Suzhou, China) were co-transfected into 293T cells with lipofectamine 2000. Cells were collected after transfection for 48 h and the effect of miRNA on luciferase activity in cells was detected by Duo-Lite^TM^ Luciferase Assay System (Vazyme Biotech Co., Ltd., Nanjing, China). The value of firefly luciferase Ff activity/renin luciferase Rn activity was used as luciferase activity.

### 2.8. RNAi and E. coli F18ac Stimulation Experiment

The target genes of miR-215 were interfered by RNAi technology. First, the single-stranded (ss) DNA oligo (Appendix A) containing the interference sequence was synthesized, and then the dsDNA oligo was generated by annealing extension. The pGMLV-SC5 vector (Genomeditech, Shanghai, China) was digested by *BamH* I and *EcoR* I, and then was recombined with dsDNA oligo by T4 ligase. The ligation product was transformed into competent cell DH-5α, and the grown monoclonal colony was sequenced and identified to construct a recombinant vector. The interference efficiency was detected by qPCR and Western blot. The interfering cells and control cells of three target genes were stimulated by *E. coli* F18ac [23]. The expression changes of related genes were detected by qPCR and the secretion levels of proinflammatory cytokines (TNF-α and IL-1β) in the cell culture medium were detected by ELISA kits (Nanjing Jiancheng Bioengineering Institute, Nanjing, China).

### 2.9. Promoter Regions Prediction and Core Promoter Identification of miR-215 Target Genes

The 2000 bp sequences upstream of the transcription start site (TSS) of porcine miR-215 target genes *EREG*, *NIPAL1* and *PTPRU* were used as templates. The sequence was truncated at the 3′ end by designing different primers (Appendix A) to amplify different promoter segments. The PCR amplification products were detected by electrophoresis and verified by sequencing. The confirmed PCR products were recovered and purified, then the pCpGL-basic vector was used to construct recombinant vectors according to the method described in our previous study [30]. 293T cells were cultured in 12-well plates with DMEM containing 10% FBS in a 37 °C incubator with 5% CO_2_. After the cell density was about 80%, the recombinant vectors were cotransfected to cells with the Renilla luciferase vector pRL-TK, and the dual luciferase assay was performed 48 h later. The procedure was strictly in accordance with the instructions of Duo-Lite^TM^ Luciferase Assay System (Vazyme Biotech Co., Ltd., Nanjing, China). At the same time, the core promoter regions of the miR-215 target genes were predicted by BDGP software (http://www.fruitfly.org/seq_tools/promoter.html).

### 2.10. Promoter Regions Prediction and Core Promoter Identification of miR-215 Target Genes

The CpG islands of promoter regions of porcine miR-215 target genes were predicted using MethPrimer 1.0 (Peking Union Medical College Hospital, Beijing, China) (http://www.urogene.org/cgi-bin/methprimer/methprimer.cgi), and the predicted fragment length was 2000 bp. The prediction parameters were set such that the sequence length of the CpG island was at least 100 bp, the GC content was greater than 50% and the CpGo/e was greater than 0.6. The duodenum tissue DNA of Sutai resistant and sensitive individuals was used as experimental material, and sulfite conversion of DNA was carried out using the EZ DNA Methylation-Gold™ Kit (ZYMO RESEARCH, Los Angeles, CA, USA). PCR primers (Appendix A) were designed using MethPrimer software and PCR amplification was performed using a ZymoTaq PreMix PCR kit (ZYMO RESEARCH, Los Angeles, CA, USA). The PCR amplification product was identified by 1% agarose gel and purified by a gelation recovery kit (Beijing Tiangen Biotechnology Co., Ltd., Beijing, China). The purified product was ligated to a pMD™ 19-T Vector (Takara Biomedical Technology (Beijing) Co., Ltd., China). The ligation product was transformed by competent-cell DH-5α and cultured. Twenty positive clones were picked from each sample and sent to Sangon Biotech Co., Ltd. (Shanghai, China) for sequencing. The DNA sequences from the genomic original sequence, sulfite conversion and amplified by methylation primers were aligned to find the CG site where methylation occurs. Correlation analysis was performed between the degree of methylation of the single site and mRNA expression.

### 2.11. Construction of Mutant Vectors of Core Promoter Regions and Detection of Dual Luciferase Activity

For the confirmed core promoter region, the mutation sites of core promoter regions of miR-215 target genes were searched using an online database (http://asia.ensembl.org/index.html). The wild-type and mutant oligos (Appendix A) of the core promoter regions were synthesized by Sangon Biotech Co., Ltd. (Shanghai, China). Oligos were denatured, annealed and extended, and then were recombined into the luciferase vector pGL3-Basic. The effects of different variant sites on promoter activities were analyzed by the Duo-Lite^TM^ Luciferase Assay System (Vazyme Biotech Co.,Ltd., Nanjing, China).

### 2.12. qPCR and Western Blot Analysis

Total RNA was extracted by Trizol reagent. The purity and concentration of total RNA were detected by 1% formaldehyde denaturing gel electrophoresis and NanoDrop-1000 micronuclei tester, and stored at −70 °C for use. cDNA synthesis and qPCR were performed strictly following the instructions of the reverse transcription kit and fluorescence quantitative detection kit (Beijing Tiangen Biotechnology Co., Ltd., Beijing, China) The GAPDH gene was used as a house-keeping gene and the relative expression level was calculated using the 2^−ΔΔCt^ method [24]. Western blot was used to detect protein expression levels of EREG, PTPRU and NIPAL1. Total protein was extracted using Trizol reagent in strict accordance with the instructions, and protein levels were normalized using a BCA kit (Thermo Fisher Scientific, Walsham, MA, USA). The conditions of sodium dodecyl sulfate polyacrylamide gel electrophoresis (SDS-PAGE) were as follows: 10 μL of the protein sample was applied to a 10% concentration gel and electrophoresed at 120 V for 90 min. Blotting: protein transfer to a polyvinylidene fluoride (PVDF) membrane and immunoblotting with related antibodies. The blocking solution and the appropriate amount of primary antibody EREG (1:600), NIPAL1 (1:400), PTPRU (1:400) and β-actin antibody (1:4000) were added in an amount of about 0.1 mL/cm^2^. Horseradish peroxidase-labeled antibody (HRP, 1:5000) was used as a secondary antibody and β-actin protein was used as a reference protein.

### 2.13. Statistical Analysis

Independent sample t test on SPSS16.0 software was applied to analyze the differences of miR-215 transcript levels in IPEC-J2 cells between the *E.coli* F18ac stimulation group and NC group, the differences of miR-215 transcript levels in IPEC-J2 cells between the miR-215 inhibit group and NC group, the differences of gens (*EREG*, *NIPAL* and *PTPRU*) transcript levels in the duodenum of *E.coli* F18-resistant and sensitive Sutai weaned piglets and the differences of gens (*EREG*, *NIPAL* and *PTPRU*) transcript levels in IPEC-J2 cells between RNAi group and NC group. One-way analysis of variance (ANOVA) on SPSS16.0 software was used to analyze the data from dual luciferase detection and the expression changes of three genes (*EREG*, *NIPAL* and *PTPRU*) in IPEC-J2 cells stimulated with the *E.coli* F18ac strain. The multivariate statistical method of general linear model (GLM) on SPSS16.0 software was applied to analyze the data from the expression changes of three genes (*EREG*, *NIPAL* and *PTPRU*) and cytokine levels in the RNAi group and NC group IPEC-J2 cells stimulated with *E.coli* F18ac bacterial culture supernatants. Three replicates were set for each treatment and the results were showed as mean ± standard deviation. The significance of the difference between the averages was tested by the LSD method. The level of statistical significance was set at *p* < 0.05. The graphs were drawn using GraphPad Prism 6.

## 3. Results

### 3.1. Expression Changes of miR-215 in IPEC-J2 Cells Stimulated by E. coli F18ac and Construction of Cell Line with miR-215 Stable Interference

The overall flow of this study is shown in Figure 1. As shown in Figure 2A, the transcription level of miR-215 in IPEC-J2 cells stimulated by *E. coli* F18ac was significantly lower than that in no-stimulation cells (*p* < 0.01). The expression of green fluorescent protein was greater than 90% after cells were screened by puromycin for 48 h, indicating that the lentiviral vectors had been successfully integrated into IPEC-J2 cells (Figure 2B). By detecting the transcription level of miR-215 in IPEC-J2 cells, the mRNA expression level of the miR-215 inhibitor group had a 0.2-fold change compared to that of the NC group (*p* < 0.01) (Figure 2C). These results indicated that the IPEC-J2 cell line with miR-215 stable interference was successfully constructed and could be used for further studies.

### 3.2. RNA-Seq Results of IPEC-J2 Cells between miR-215 Inhibitor Group and NC Group

In this study, RNA-seq technology was used to obtain transcripts of IPEC-J2 cells with miR-215 inhibitor group and NC group. The clean data of the sequencing results were uploaded to the NCBI database (Accession number: PRJNA445990, PRJNA5997, PRJNA5998, PRJNA5999, PRJNA6000 and PRJNA6004). There were 352 differentially expressed genes (DEGs) found between the miR-215 inhibitor group and NC group (L vs. NC), of which 302 were up-regulated and 50 were down-regulated (Figure 3A). Go function results showed that functional annotation of DEGs were mainly involved in biological processes (localization and transport), cellular components (membrane part and membrane region) and molecular function (transporter activity and adenyl nucleotide binding) (Figure 3B). KEGG pathway results showed that the DEGs were involved in the pathways of adherens junction, bacterial invasion of epithelial cells and the epidermal growth factor receptor (EGFR, ErbB) signaling pathway (Figure 3C). Partial DEGs were shown in Figure 3D; qPCR validation analysis revealed that the expression trends of DEGs were consistent with the DEGs identified by transcriptome sequencing (Figure 3E).

### 3.3. Prediction and Verification of miR-215 Target Genes

Combined with the DEGs of sequencing and the predicted results, the candidate target genes for miR-215 were initially screened as *EREG*, *NIPAL1* and *PTPRU*. The recombinant plasmids pMIR-Report of miR-215 target genes were constructed and validated by Sanger sequencing (Appendix A). The results showed that miRNA-215 mimics showed significant inhibition on *EREG* and *NIPAL1* (*p* < 0.05), and showed significant inhibition on *PTPRU* (*p* < 0.01); miRNA-215 inhibitor showed significant enhancement for *EREG* and *NIPAL1* (*p* < 0.05), and showed significant enhancement for *PTPRU* (*p* < 0.01) (Figure 4A).

qPCR was used to detect the expression of miR-215 target genes (*EREG*, *NIPAL1* and *PTPRU*) in the duodenum of *E.coli* F18-resistant and sensitive Sutai weaned piglets. The results showed that the expression levels of *EREG* and *PTPRU* genes in resistant individuals were significantly higher than those in sensitive individuals (*p* < 0.05), and the expression level of the *NIPAL1* gene in resistant individuals was significantly higher than that in sensitive individuals (*p* < 0.01) (Figure 4B). The results of correlation analysis between the expression level of miR-215 in the duodenum of *E.coli* F18-resistant and sensitive Sutai weaned piglets, and the expression of three target genes revealed that the expression level of miR-215 showed a significant negative correlation with the expression level of the three target genes (*p* < 0.05) (Figure 4C–E).

### 3.4. The Regulation of miR-215 Target Genes Expression Levels in the Process of E. coli Infection of IPEC-J2 Cells

The interference vectors pGMLV-EREG, pGMLV-NIPAL and pGMLV-PTPRU of the miR-215 target genes (*EREG*, *NIPAL1* and *PTPRU*) were obtained by RNAi technology (Appendix A). The results showed that the expression levels of *EREG*, *NIPAL1* and *PTPRU* were significantly down-regulated (*p* < 0.01), and protein expression levels were also down-regulated (Figure 5A,B).

*E.coli* F18ac strain and bacterial culture supernatants were used to stimulate IPEC-J2 cells with the *EREG*, *NIPAL1* and *PTPRU* RNAi group and the negative control group (pGMLV-NC) to detect the expression changes of three genes (Figure 5C). The results showed that the expression levels of *EREG*, *NIPAL1* and *PTPRU* genes in the pGMLV-NC group were significantly up-regulated after bacterial stimulation (*p* < 0.01), and the expression levels of the three target genes showed a continuous upward trend with increased stimulation time. However, the expression levels of *EREG*, *NIPAL1* and *PTPRU* genes did not change significantly in the RNAi group (*p* > 0.05) and were significantly lower than in the control group (*p* < 0.05 or *p* < 0.01).

The levels of proinflammatory cytokines TNF-α and IL-1β in IPEC-J2 cells were measured by ELISA and a standard curve was established (Appendix A). The results of ELISA showed that the levels of proinflammatory cytokines TNF-α and IL-1β in the RNAi group and pGMLV-NC group increased with stimulation time (Figure 5D). Of importance was the levels of TNF-α and IL-1β in the RNAi group were significantly higher than those in the pGMLV-NC group (*p* < 0.05 or *p* < 0.01).

### 3.5. Detection of Promoter Methylation in Target Genes of miRNA-215 and Effect of SNPs in Core Promoter Region on Promoter Activity

As shown in Figure 6A, there was one CpG island in the promoter region of the *NIPAL1* gene, which contained five CG sites (Figure 6A). There was one CpG island in the promoter region of the *PTPRU* gene, which contained four CG sites (Figure 6B). However, no CpG island was predicted in the promoter region of the *EREG* gene (Figure 6C). Methylation of the single CG site of the *NIPAL1* gene and *PTPRU* gene promoter regions in the duodenum between *E.coli* F18ac-resistant and sensitive Sutai weaned piglets was detected, respectively. The results showed that the methylation degree of the *NIPAL1* gene was very low (≤10%), while the methylation degree of the *PTPRU* gene was relatively high (≥50%) (Figure 6D,E). Correlation analysis between the methylation degree and mRNA expression level revealed that methylation at the mC1 site of the *NIPAL1* gene was positively correlated with expression level, but there was no significant difference (*p* > 0.05) (Figure 6F). The methylation at the mC1 and mC2 sites of the *PTPRU* gene was negatively correlated with expression level, the mC3 site was positively correlated with the expression level, but there was no significant difference (*p* > 0.05) (Figure 6G).

Dual luciferase assays were performed on different segments of the promoter regions of the miR-215 target genes (Figure 7A). There were two positive regulatory elements located at 706–1172 bp and 53–102 bp and a negative regulatory element located at 274–706 bp upstream of the transcription start site (TSS) in the *NIPAL1* gene. There was a positive regulatory element located at 800–1038 bp upstream of the TSS in the *EREG* gene. There was a positive regulatory element located at 884–1107 bp upstream of the TSS in the *PTPRU* gene. Combining the core promoter region and SNP database information, the wild type and mutant luciferase vectors of core promoter for miR-215 target genes were constructed separately (Appendix A). Luciferase activity assay results showed that (Figure 7B) there were seven SNPs in the *NIPAL1* gene, and the dual luciferase activities of SNP1, 2, 4 and 7 sites were significantly decreased (*p* < 0.01). The dual luciferase activity of the SNP5 site was significantly increased (*p* < 0.05). It was predicted by website that the SNP2 site of the *NIPAL1* gene was located at the binding domain of transcription factor RAP1, and the SNP5 variable site was located at the binding domain of transcription factor SP1. There was only one SNP in the *EREG* gene, and the dual luciferase activity of the variable site was significantly increased (*p* < 0.01) (Figure 7C). There was an insertion mutation and a point mutation in the *PTPRU* gene, and the dual luciferase activity of insertion mutation site was significantly decreased (*p* < 0.01) (Figure 7D).

## 4. Discussion

In this study, we first verified the relationship between the expression of miR-215 and *E. coli* stimulation in IPEC-J2 cells and found that the expression level of miR-215 was significantly down-regulated in *E. coli* F18ac-stimulated IPEC-J2 cells, which suggested that miR-215 may play an important regulatory role in the process of *E. coli* infection. miRNAs indirectly affect phenotype and function by regulating the expression of target genes. Therefore, we knocked down the endogenous expression of miR-215 in IPEC-J2 cells using an miR-215 inhibitor lentivirus and then performed transcriptome sequencing analysis of IPEC-J2 cells with the miR-215 inhibitor group and NC group. Combined analysis of the differentially expressed genes with the predicted target genes and three candidate target genes, *EREG*, *NIPAL1* and *PTPRU*, were finally determined. Through the detection of dual luciferase activity and expression correlation analysis with miR-215, we further determined that miR-215 had a targeted inhibitory effect on *EREG*, *NIPAL1* and *PTPRU* genes.

Epiregulin (EREG) belongs to the epidermal growth factor (EGF) family, and is initially expressed as a transmembrane protein and secreted extracellularly in a soluble form upon maturity, which is an autocrine or paracrine growth factor [31]. EREG functions by specific binding to its receptor EGFR (ErbB), which causes the activation of tyrosine kinases and ultimately regulates cell activity [32]. Studies have shown that the expression of EREG was associated with various tumor diseases such as lung cancer, gastric cancer and colon cancer, which revealed that EREG played an important role in the occurrence and development of tumors [33,34,35]. Recent studies revealed that porcine *EREG* played an important role in oocyte development and maturation, and the expression of *EREG* enhanced the repair of wounds [36,37,38,39]. The protein tyrosine phosphatase receptor U (PTPRU) is a receptor-type tyrosine pepsin (PTP), which is a member of the R2B receptor PTPRU subfamily. The PTPRU protein consists of a mitochondria associated membrance (MAM) domain, an Ig-like domain, three consecutive Fibronectin III (FNIII) domains, and two intracellular phosphatase domains from the N-terminus to the C-terminus. PTPRU was involved in maintaining epithelial integrity, regulating Wnt/β-catenin signaling pathways and neurodevelopment and has been shown to act as a regulator of adhesion and proliferation in colon, breast, glioma and gastric cancer [40,41,42,43,44,45]. A NIPA like domain containing 1 (NIPAL1) encodes a magnesium transporter, which is associated with recurrent oral squamous cell carcinoma, colon cancer, gout and neurodevelopment [40,46,47,48]. At present, there are few reports on pig *PTPRU* and *NIPAL1* genes. This study is expected to reveal their new functional annotations in pigs.

The results of this study showed that the expression levels of *EREG*, *NIPAL1* and *PTPRU* genes in *E. coli* F18-resistant individuals were significantly higher than those in sensitive individuals and the expression levels of these three genes were significantly negatively correlated with the expression level of miR-215. These results not only revealed that these three genes were indeed related to the resistance of *E. coli*, but also showed their targeted relationship with miR-215. Thuong et al., used a genome-wide association study and found that the polymorphisms of the *EREG* gene are associated with tuberculosis (TB) susceptibility [49]. It was reported that in the setting of pretreated advanced colorectal cancer (ACRC), patients with high *EREG* gene expression may benefit from cetuximab therapy [50]. Therefore, this study suggested that high expression of the *EREG* gene may be beneficial to piglets’ resistance to *E. coli*. Previous research reported that miR-574-5p enhanced tyrosine phosphorylation of β-catenin by repressing *PTPRU* expression in vitro and promotes the migration and invasion of nonsmall cell lung cancer (NSCLC) cells [51]. The results of this study indicated that miR-215 could regulate *E. coli* infection by targeting *EREG*, *NIPAL1* and *PTPRU* genes.

Promoter methylation is one of the main ways of epigenetic modification. Numerous studies have shown that promoter methylation plays an important role in gene expression, growth and development disease and immunity [52,53,54]. Therefore, promoter methylation of these three genes were detected in this study. It was found that there was one CpG island in the promoter regions of *NIPAL1* and *PTPRU* genes but no CpG island was predicted in the *EREG* gene promoter region. Correlation analysis showed that there was no significant correlation between methylation and mRNA expression levels of CpG islands in the promoter regions of *NIPAL1* and *PTPRU* genes. It has been reported that the methylation of the human *EREG* gene promoter region can affect its expression level [35,55]. These results indicated that expression of *EREG*, *NIPAL1* and *PTPRU* genes may not be regulated by promoter methylation.

Genetic variation in the promoter region could also affect gene expression [56,57,58]. The results showed that the site mutation in the promoter of the *EREG* gene could induce the enhancement of transcriptional activity and could be further analyzed as a molecular marker. An insertion mutation site in the promoter of the *PTPRU* gene could cause a decrease in transcriptional activity. The dual luciferase activity of the variant sites SNP1, 2, 4 and 7 of the *NIPAL1* gene was significantly reduced. It was predicted that SNP2 of the *NIPAL1* gene was located in the binding domain of the transcription factor RAP1. Rap1, a transcription factor, binds to most yeast ribosomal protein promoters (RPs) and the Rap1 site is important for the coordinated regulation of the RP gene [59]. We hypothesized that the transcription factor Rap1 may enhance the transcription activity of the *NIPAL1* gene in this study. The dual luciferase activity at SNP5 of the *NIPAL1* gene was significantly increased, and this site was located in the binding domain of transcription factor SP1. SP1 is a ubiquitously expressed prototype C2H2-type zinc-binding DNA binding protein, which can activate or inhibit transcription in response to physiological and pathological stimuli [60]. We hypothesized that SP1 may inhibit the transcription activity of the *NIPAL1* gene in this study. Based on the above results, both sites of the *NIPAL1* gene could be further studied as molecular markers.

## 5. Conclusions

In conclusion, this study illustrated that porcine miR-215 could target *EREG*, *NIPAL1* and *PTPRU* genes to regulate the resistance of piglets to *E. coli* F18 (Figure 7). Although results in this study indicated that expression of *EREG*, *NIPAL1* and *PTPRU* genes may not be regulated by promoter methylation. There were SNPs in the core promoter regions of *EREG*, *NIPAL1* and *PTPRU* genes that could regulate their transcriptional activity, and the *EREG* gene mutation site, the *PTPRU* gene insertion mutation site and the *NIPAL1* gene mutation sites could be further studied as molecular markers. Importantly, this study revealed new biological functions of *EREG*, *NIPAL1* and *PTPRU* genes in pigs for the first time, which may contribute to further research of disease-resistance breeding in pigs.

## Figures and Tables

**Figure 1 genes-11-01053-f001:**
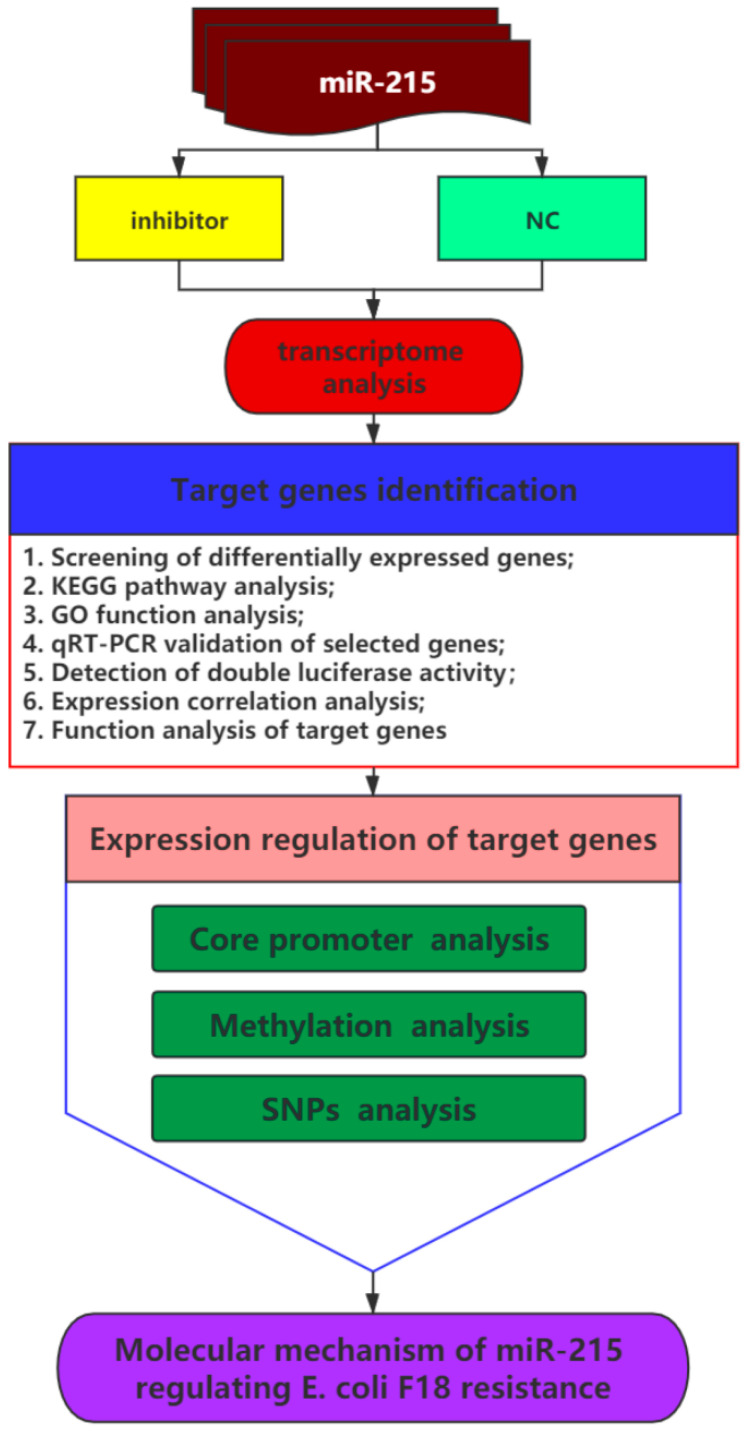
The overall flow of this study (Abbreviations: NC, negative control; KEGG, Kyoto Encyclopedia of Genes and Genomes; GO, Gene Ontology; qRT-PCR, real-time quantitative PCR; SNPs, single nucleotide polymorphisms; *E. coli*, *Escherichia coli*).

**Figure 2 genes-11-01053-f002:**
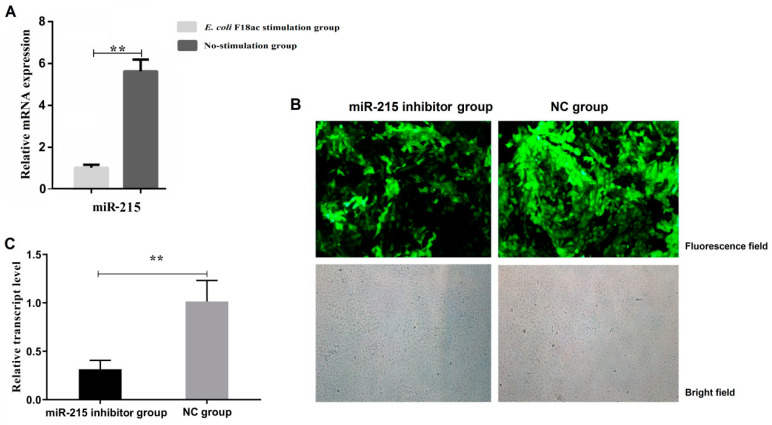
Expression changes of miR-215 in IPEC-J2 cells stimulated by *E. coli* F18ac and construction of cell line with miR-215 stable interference. (**A**) Transcription level of miR-215 in IPEC-J2 cells between the *E. coli* F18ac stimulation group and no-stimulation group. (**B**) Expression of green fluorescent protein after IPEC-J2 cells were transfected by lentivirus for 48 h (100×). (**C**) Transcription level of miR-215 after IPEC-J2 cells were transfected by lentivirus for 72 h. ** indicates the significant difference (*p* < 0.01).

**Figure 3 genes-11-01053-f003:**
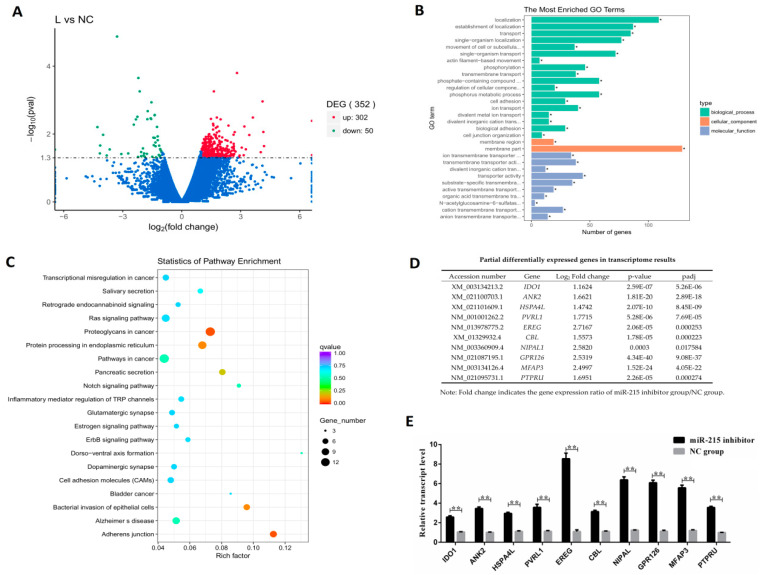
Transcriptome sequencing results and qPCR verification of partial differential genes. (**A**) Volcano plot of differentially expressed genes (DEGs) between miR-215 inhibitor group and NC group. Genes with significant differential expression are indicated by red dots (up-regulation) and green dots (down-regulation), and genes with no significant differential expression are indicated by blue dots. The abscissa represents the fold-change of gene expression in different samples. The ordinate represents the statistical significance of the difference in the amount of change in gene expression. L, miR-215 inhibitor group; NC, negative control group. (**B**) GO enrichment histogram. The ordinate shows the enriched GO term and the abscissa shows the number of differential genes in the term. Different colors are used to distinguish biological processes, cellular components, and molecular functions; * indicates significantly enriched GO term. (**C**) KEGG enrichment scatter plot of differential genes. The vertical axis shows pathway names and the horizontal axis shows rich factor. The size of the dots indicates the number of differential genes in the pathway, and the color of the dots corresponds to a different Qvalue range. (**D**) Partial differentially expressed gene (DEGs) in transcriptome results. (**E**) Differentially expressed genes (DEGs) detected by qPCR. ** indicates significant difference (*p* < 0.01).

**Figure 4 genes-11-01053-f004:**
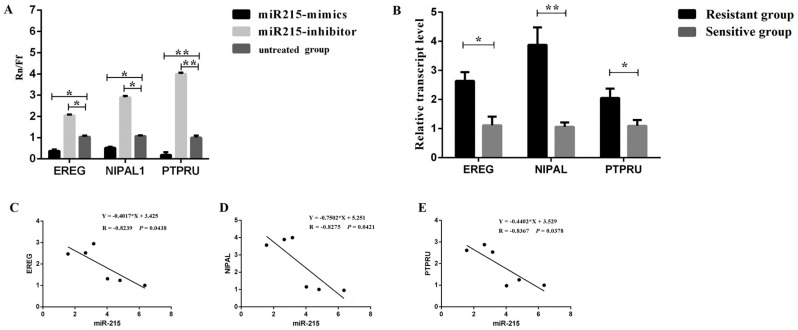
Verification of the targeting relationship between miR-215 and its target genes. (**A**) Detection of dual luciferase activity of target genes. (**B**) Expression of target genes in the duodenum between *E. coli* F18-resistant and sensitive Sutai weaned piglets. (**C**–**E**) Expression correlation analysis of miR-215 and target genes in which the expression data of miR-215 was derived from our previous study results [21]. *, *p* < 0.05; **, *p* < 0.01.

**Figure 5 genes-11-01053-f005:**
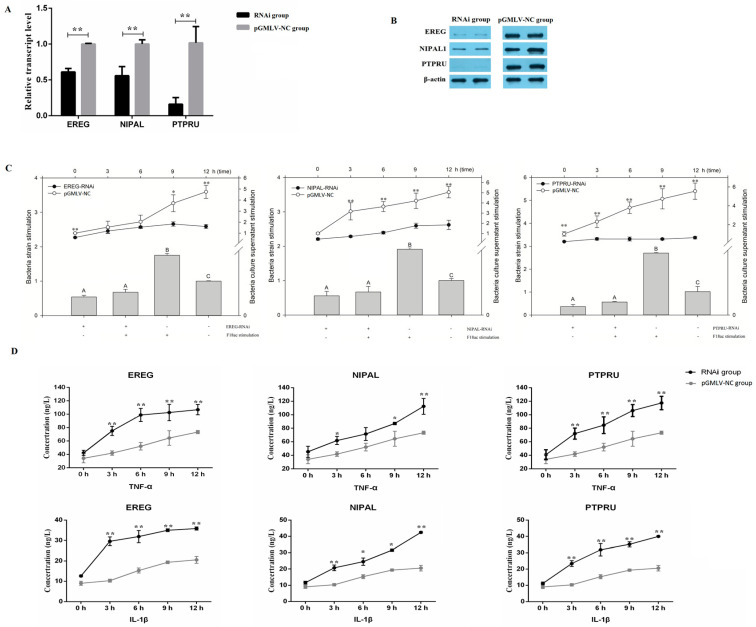
Effect of miR-215 target genes expression levels on *E. coli* F18ac infection of IPEC-J2 cells. Expression level changes in mRNA (**A**) and protein (**B**) of miR-215 target genes were detected after IPEC-J2 cells were transfected by interference vectors. (**C**) The mRNA expression changes of miR-215 target genes after IPEC-J2 cells were stimulated by *E. coli* F18ac strain and bacterial supernatants at different time points (0, 3, 6, 9 and 12 h). (**D**) Proinflammatory cytokine levels after IPEC-J2 cells were stimulated by bacterial supernatants at different time points. *, *p* < 0.05; **, *p* < 0.01.

**Figure 6 genes-11-01053-f006:**
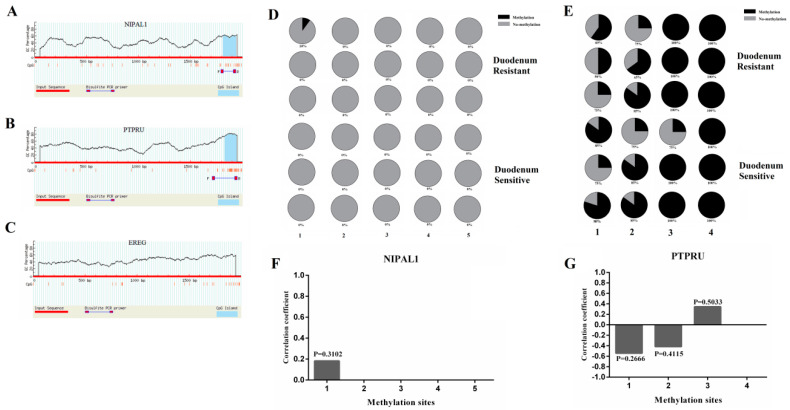
CpG islands prediction and methylation detection in promoter regions of miR-215 target genes. (**A**–**C**) CpG island prediction in promoter regions of miR-215 target genes *NIPAL1*, *PTPRU* and the *EREG* gene, respectively. (**D**) and (**E**) Methylation degree in promoter regions of the *NIPAL1* and *PTPRU* gene in F18 resistant and sensitive duodenum of Sutai piglets, respectively. (**F**) and (**G**) Correlation between methylation degree and mRNA expression of the *NIPAL1* and *PTPRU* gene, respectively.

**Figure 7 genes-11-01053-f007:**
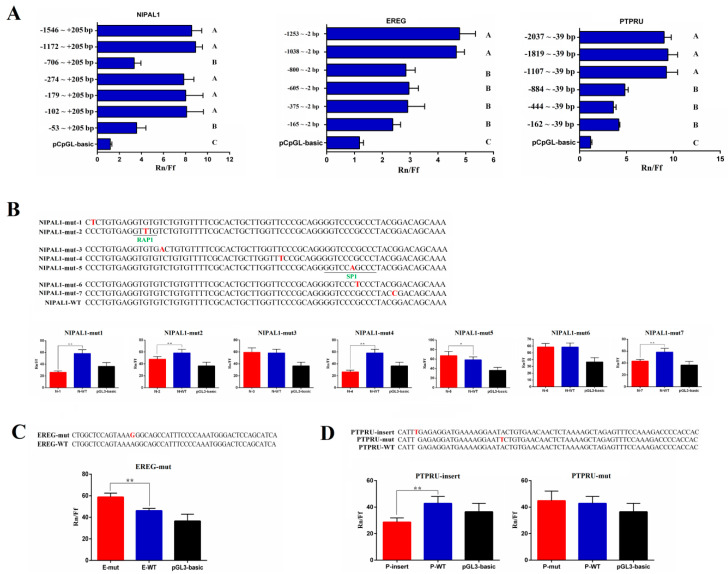
Identification of the core promoter regions of miR-215 target genes and the effects of SNPs on promoter activities. (**A**) Dual luciferase activity in different segments of promoters of the miR-215 target genes *NIPAL1*, *EREG* and *PTPRU* genes. (**B**–**D**) Wild type and mutant sequences of *NIPAL1*, *EREG* and *PTPRU* genes core promoters and their effects on promoter activities. Red letters represent mutation sites, green letters and underlines represent transcription factors and their binding sites.

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
