# Peer review of "miR-215 Targeting Novel Genes *EREG*, *NIPAL*1 and *PTPRU* Regulates the Resistance to *E.coli* F18 in Piglets"

_genes, 2020, doi:10.3390/genes11091053_

Round 1

Reviewer 1 Report

In this study, the authors used RNA-seq technology to the potential function of miR-215 in porcine. The results can be used to improve the annotation of some novel genes in this poorly annotated genome. Some minor comments:

Line 146- please add a sentence to describe the definition of “clean reads”

Line 151, please add a few words describing what “padj” imply to. Also, please mention what type of multiple-test adjustment method used in this study.

Line 151, please add the software/package used for genes functional enrichment analysis and the set of genes used as background in this analysis.

Author Response

In this study, the authors used RNA-seq technology to the potential function of miR-215 in porcine. The results can be used to improve the annotation of some novel genes in this poorly annotated genome. Some minor comments:

 Line 146- please add a sentence to describe the definition of “clean reads”

Thanks for your suggestion. We added the definition of “clean reads”to line 140 “ (clean data were obtained by removing reads containing adapter, reads containing ploy-N and low quality reads from raw data.)” in revised manuscript.

Line 151, please add a few words describing what “padj” imply to. Also, please mention what type of multiple-test adjustment method used in this study.

Thanks for your advice. We added the description to line 147 (Here, “padj” refers to “p-adjusted” that is corrected after multiple-test adjustment by the method of Benjamini& Hochberg.) in revised manuscript.

Line 151, please add the software/package used for genes functional enrichment analysis and the set of genes used as background in this analysis.

Thanks for your comments. We modified this part in line 149 “GO seq based Wallenius non-central hyper-geometric distribution [28] was implemented for GO enrichment analysis. KOBAS software [29] was used to test the statistical enrichment of the differentially expressed genes in KEGG pathways. All the gene sets annotated to the Entrez database for significant difference analysis were used as background in this analysis.” in revised manuscript.

In addition, we improved the methods part to make the manuscript more concise.

Reviewer 2 Report

  1. Original Submission
    • Recommendation

Minor Revision

  1. Comments to Author:

Ref. No.: genes-889956

miR-215 targeting Novel Genes EREG, NIPAL1 and PTPRU Regulates the Resistance to E.coli F18 in Piglets

Chaohui Dai , Fang Wang , Shiqin Wang , Zhengchang Wu , Shenglong Wu , Wenbin Bao

Overview and general recommendation:

A microRNA is a small non-coding RNA molecule that functions in RNA silencing and post-transcriptional regulation of gene expression. miRNAs are widely involved in various organ development, cell proliferation and differentiation, migration and apoptosis, lipogenesis metabolism, insulin secretion and other physiological activities. The work aimed to review annotated new biological functions of EREG, NIPAL1 and PTPRU genes in pigs, which provided a new experimental basis and reference for the research of piglets disease-resistance breeding. All three genes were “controlled” by the mir-215.

It is one of the highly complex studies that I had to deal with. The authors consider a few steps thorough the RNASeq method; retargeting for miR genes; RNAi; promotor regions identification; epigenetics and finally the luciferase activity. The particular attention was being paid to a detailed description of all the laboratory methods used in the experiment. However, it sometimes right to say less than more and the lab methods part should be rewritten to a more strict form. Some parts of the methodology could be transferred to the supplementary section. And on the other hand, the statistical and bioinformatic part was described less than it should.

Moreover, the authors cite a lot of work from 2012:

Ye L, Su X, Wu Z, Zheng X, Wang J, Zi C, et al. (2012) Analysis of Differential miRNA Expression in the Duodenum of Escherichia coli F18-Sensitive and -Resistant Weaned Piglets. PLoS ONE 7(8): e43741. https://doi.org/10.1371/journal.pone.0043741

What was the reason that the authors back to the subject after right years?

It is hard to understand why the authors used from the whole spectrum of genes (903 up-regulated and 251 down-regulated) only this three: EREG, NIPAL1 and PTPRU. Looking at the short table (Figure 2D), there was no durable feel that these genes should be selected as potential candidates (NIPAL1 high p-value, PTPRU low log2 FC).

The introduction part is clear. However, there was no full literature review in the aspect of the miRs and pigs in the example there was no paper:

Ropka-Molik, K.; Pawlina-Tyszko, K.; Żukowski, K.; Tyra, M.; Derebecka, N.; Wesoły, J.; Szmatoła, T.; Piórkowska, K. Identification of Molecular Mechanisms Related to Pig Fatness at the Transcriptome and miRNAome Levels. Genes 2020, 11, 600. https://www.mdpi.com/2073-4425/11/6/600

And the authors mainly focused on the relevant regulatory roles of miRs in pathological processes such as a tumour, viral and bacterial diseases. The results and discussion sections were well-constructed and described, and with the conclusion section were mainly focused on the description of the EREG, NIPAL1 and PTPRU genes.

The research was based on the ethics statement. Moreover, there was a publication of dataset in public databases (NCBI).

Therefore, I recommend that a minor revision is warranted. I explain my concerns in more detail below. I ask that the authors explicitly address each of my comments in their response.

Additional comments:

  1. Why did the authors decide to use such a low fold change level >0.5? Moreover, looking on the Figure 2D – pvalue for NIPAL1 gene was 0.0003, and this gene did not pass the padj level, should be removed from the analysis. Is it true or error in Figure 2D (padj instead of pvalue). Could the authors add the full list of genes in csv format at the Supplementary materials?
  2. Is the experimental design (three interference biological samples (L1-L3) vs three control samples (NC1-NC3) was only for the RNASeq part and was transferred for the whole study or it was exclusively designed for this part of the analysis?
  3. Is there could be added some diagram of the complete analysis?
  4. The improvement in English is needed; the text is too worldly. A lot of double-stranded/single-stranded DNA…

Author Response

Overview and general recommendation:

It is one of the highly complex studies that I had to deal with. The authors consider a few steps thorough the RNASeq method; retargeting for miR genes; RNAi; promotor regions identification; epigenetics and finally the luciferase activity. The particular attention was being paid to a detailed description of all the laboratory methods used in the experiment. However, it sometimes right to say less than more and the lab methods part should be rewritten to a more strict form. Some parts of the methodology could be transferred to the supplementary section. And on the other hand, the statistical and bioinformatic part was described less than it should.

Thanks for your comments. We noticed that some methods were performed according to the relative kits instructions and some experimental description could be cited from previous studies, so we improved the methods part to make the manuscript more concise. In addition, we added more details in the statistical part (line 240) and bioinformatic part (line 140, 143 and 147).

Moreover, the authors cite a lot of work from 2012:

Ye L, Su X, Wu Z, Zheng X, Wang J, Zi C, et al. (2012) Analysis of Differential miRNA Expression in the Duodenum of Escherichia coli F18-Sensitive and -Resistant Weaned Piglets. PLoS ONE 7(8): e43741. https://doi.org/10.1371/journal.pone.0043741

What was the reason that the authors back to the subject after right years?

Thanks for your comments. As you said, we cited the reference (Ye et al. 2012) for 3 times in our manuscript (at the end of the first paragraph of the introduction part, the expression data of miR-215 in figure 3 and the first sentence of the discussion part, respectively). We cited it in introduction because it provided the basis of this study, in which miR-215 was a differentially expressed miRNA in the duodenum between E. coli F18 resistant and sensitive Sutai pigs. We hypothesized that miR-215 might be an important miRNA that regulates piglets’ resistance to E. coli F18. However, the molecular mechanism of miR-215 regulating the resistance of piglets to E. coli F18 remains poorly understood. Hence, we aimed to study the potential function and molecular mechanism of miR-215. We cited the expression data of miR-215 in Figure 3 because we wanted to analyze the expression correlation analysis of miR-215 and target genes (EREG, NIPAL1 and PTPRU) to further confirm their target relationship. We conducted the expression of EREG, NIPAL1 and PTPRU genes in the duodenum between E. coli F18 resistant and sensitive Sutai weaned piglets in this study, and the expression of miR-215 in the duodenum between E. coli F18 resistant and sensitive Sutai weaned piglets were conducted by Ye et al. They were both from the same duodenum tissues and detected by qPCR. Therefore, we cited the data of miR-215 expression rather than conducted the experiments again. As for the reference in the discussion, we deleted it because it didn't seem to be so important in this part.

It is hard to understand why the authors used from the whole spectrum of genes (903 up-regulated and 251 down-regulated) only this three: EREG, NIPAL1 and PTPRU. Looking at the short table (Figure 2D), there was no durable feel that these genes should be selected as potential candidates (NIPAL1 high p-value, PTPRU low log2 FC).

Thanks for your comments. We agree that these three genes (EREG, NIPAL1 and PTPRU) were not the most significant differentially expressed genes in RNA-seq results. However, we found that miR-215 could target these three genes best when we predicted the target genes of miR-215 on the online websites TargetScan and miRTarBase. Combined the DEGs of sequencing with the predicted results, we selected these three genes as potential candidates. Furthermore, we verified the transcript levels differences by qPCR and the important functions of these genes in this study. miR-215 truly targeted these three genes and regulated the the resistance to E.coli F18.

The introduction part is clear. However, there was no full literature review in the aspect of the miRs and pigs in the example there was no paper:

Ropka-Molik, K.; Pawlina-Tyszko, K.; Żukowski, K.; Tyra, M.; Derebecka, N.; Wesoły, J.; Szmatoła, T.; Piórkowska, K. Identification of Molecular Mechanisms Related to Pig Fatness at the Transcriptome and miRNAome Levels. Genes 2020, 11, 600. https://www.mdpi.com/2073-4425/11/6/600

Thanks for your suggestion. We added the relative reference to line 43 “Recent research identified molecular network within genes and miRNAs were candidate factors determining adipogenesis, fatness, and selected fattening characteristics in pigs [10]” in revised manuscript.

Additional comments:

Why did the authors decide to use such a low fold change level >0.5? Moreover, looking on the Figure 2D – pvalue for NIPAL1 gene was 0.0003, and this gene did not pass the padj level, should be removed from the analysis. Is it true or error in Figure 2D (padj instead of pvalue). Could the authors add the full list of genes in csv format at the Supplementary materials?

Thanks for your comments. We reanalyzed the RNA-seq data and set |log2(Foldchange)| >1 and padj < 0.05. In fact, the padj for NIPAL1 gene was 0.017584 which showed the significant difference. We added both p value and padj in revised Figure 2D. In addition, we added the full list of DEGs in csv format at the Supplementary materials.

Is the experimental design (three interference biological samples (L1-L3) vs three control samples (NC1-NC3) was only for the RNASeq part and was transferred for the whole study or it was exclusively designed for this part of the analysis?

Thanks for your comments. The experimental design (three interference biological samples (L1-L3) vs three control samples (NC1-NC3) was only for the RNA-seq and qPCR parts. To avoid confusion with the names of other cell samples, we have updated the cell groups names for different treatments (including figures) in revised manuscript.

Is there could be added some diagram of the complete analysis?

Thanks for your suggestion. We tried to add a diagram for overall flow of this study in revised manuscript (Figure 1).

The improvement in English is needed; the text is too worldly. A lot of double-stranded/single-stranded DNA…

Thanks for your comments. We rewrote the manuscript to improve English editing.
